# The Efficacy of Immunotherapy in Long-Term Survival in Non-Small Cell Lung Cancer (NSCLC) Associated with the Syndrome of Inappropriate Antidiuretic Hormone Secretion (SIADH)

**DOI:** 10.3390/life13061279

**Published:** 2023-05-30

**Authors:** Roxana-Andreea Rahnea-Nita, Alexandru-Rares Stoian, Rodica-Maricela Anghel, Laura-Florentina Rebegea, Anda-Natalia Ciuhu, Xenia-Elena Bacinschi, Anca-Florina Zgura, Oana-Gabriela Trifanescu, Radu-Valeriu Toma, Georgiana Bianca Constantin, Gabriela Rahnea-Nita

**Affiliations:** 1The Clinical Department, The Faculty of Medicine, The University of Medicine and Pharmacy “Carol Davila”, 050474 Bucharest, Romania; roxana_rahnea@yahoo.com (R.-A.R.-N.); dr.raresstoian@yahoo.com (A.-R.S.); rodicamanghel@gmail.com (R.-M.A.); xenia_bacinschi@yahoo.com (X.-E.B.); medicanca@gmail.com (A.-F.Z.); oanatrifanescugab@yahoo.com (O.-G.T.); tomaraduvaleriu@yahoo.com (R.-V.T.); gabriela_rahnea@yahoo.com (G.R.-N.); 2“Sf. Luca” Chronic Disease Hospital, 041915 Bucharest, Romania; andadum@yahoo.com; 3“Bagdasar-Arseni” Emergency Clinical Hospital, 041915 Bucharest, Romania; 4The Oncological Institute “Prof. Dr. Alexandru Trestioreanu”, 022328 Bucharest, Romania; 5The Radiotherapy Department, “Sf. Ap. Andrei” County Emergency Clinical Hospital, 800579 Galati, Romania; laura_rebegea@yahoo.com; 6The Clinical Department, The Faculty of Medicine and Pharmacy “Dunarea de Jos” University in Galati, 800008 Galati, Romania; 7The Research Center in the Field of Medical and Pharmaceutical Sciences, ReFORM-UDJ, 800010 Galati, Romania; 8The Morphological and Functional Sciences Department, The Faculty of Medicine and Pharmacy, “Dunarea de Jos” University in Galati, 800008 Galati, Romania

**Keywords:** non-small cell lung cancer, syndrome of inappropriate antidiuretic hormone secretion, immunotherapy, long-term survival

## Abstract

Introduction: The syndrome of inappropriate antidiuretic hormone secretion (SIADH) is the most common cause of hyponatremia in cancer patients, occurring most frequently in patients with small cell lung cancer. However, this syndrome occurs extremely rarely in patients with non-small cell lung cancer. The results of the clinical trials have revealed that immuno-oncological therapies are effective for long periods of time, providing hope for long survival and with a good quality of life. Case Presentation: We present the case of a female patient who was 62 years old at the time of diagnosis in 2016 who underwent surgery for a right pulmonary tumor (pulmonary adenocarcinoma) and subsequently underwent adjuvant chemotherapy. The patient had a left inoperable mediastinohilar relapse in 2018, which was treated using polychemotherapy The patient also had an occurrence of progressive metastasis and a syndrome of inappropriate antidiuretic hormone secretion (SIADH) in 2019 for which immunotherapy was initiated. The patient has continued with immunotherapy until the time this study began to be written (April 2023), the results being the remission of hyponatremia, the clinical benefits and long-term survival. Discussion: The main therapeutic option for SIADH in cancer patients is the treatment of the underlying disease, and its correction depends almost exclusively on a good response to oncological therapy. The initiation of immunotherapy at the time of severe hyponatremia occurrence led to its remission as well as the remission of the other two episodes of hyponatremia, which the patient presented throughout the evolution of the disease, demonstrating an obvious causal relationship between SIADH and the favorable response to immunotherapy. Conclusions: Each patient must be approached individually, taking into account the various particular aspects. Immunotherapy proves to be the innovative treatment that contributes to increasing the survival of patients with metastatic non-small cell lung cancer and to increasing their quality of life.

## 1. Introduction

Lung cancer is the leader in terms of incidence and mortality worldwide, with an estimated number of two million newly diagnosed cases and 1.8 million deaths. 

It is the second most frequent form of cancer after prostate cancer in men, and after breast cancer in women [1].

According to the WHO 2015, lung cancer is divided into two large categories: non-small cell lung cancer, subdivided into adenocarcinoma and squamous carcinoma; and neuroendocrine lung cancer, subdivided into microcellular cancer, large cell neuroendocrine and carcinoid tumors [2].

Smoking, both active and passive, is considered the main risk factor, being responsible for more than 80% of all cases [3]. It is followed by exposure to radon (approximately 10% and over 30% of all lung cancers in non-smokers), asbestos, air pollution, arsenic and other lung diseases [4].

Among the non-modifiable risk factors, we mention age, male gender (at least twice as often), Afro-American men from the USA and genetic factors [5].

The low survival rate of five years for non-small cell lung cancer of 64% in localized disease and 8% in metastatic disease, and 29% for lung cancer in localized disease and 3% in metastatic disease, respectively [6], makes this pathology one with a major impact on health worldwide, requiring each aspect, both general and particular, to be given special attention, representing an extremely important chapter in oncological pathology.

In the field of oncology, new treatments and evidence-based medicine have determined major progress both in terms of survival and quality of care, with a patients’ life quality a major objective in patient care [7,8].

The current paper aims at highlighting the effectiveness of immunotherapy, combining theoretical and practical notions by presenting a clinical trial in an extremely particular situation, namely the occurrence of the paraneoplastic SIADH syndrome in a female patient with non-small cell lung cancer.

The syndrome of inadequate antidiuretic hormone secretion (SIADH) is characterized by euvolemic hypotonic hyponatremia and it may be a consequence of the following factors: hyper-production of arginine vasopressin (an antidiuretic hormone – ADH) at the level of the pituitary gland as a response to a stimulus from tumor cells through the ectopic production of ASH-like peptides in the tumor cells, or as an adverse reaction to some drugs that stimulate the production of the antidiuretic hormone [9].

The diagnosis is made after ruling out other causes of hyponatremia, such as hypovolemia, hypervolemia (organ failure), hyperthyroidism and adrenal cortex failure [10].

SIADH is the most common cause of hyponatremia in cancer patients. More than 70% of the cases are attributed to microcellular lung cancer (with a frequency of 7–16% among this group of patients); while non-small cell lung cancer is attributed with a very low percentage, approximately 0.4–2% [11]. Other cancers, in which it is found, are head and neck cancers (approximately 3%), breast cancer, genitourinary cancers and sarcomata.

In oncological pathology, SIADH may be a paraneoplastic syndrome, in which case there is no direct relation with the location of the primary tumor or metastases but with the ectopic activity of the cancer cells. An optimal response to the oncological treatment can normalize the level of arginine vasopressin [12].

Moreover, certain cytostatic drugs or some other drugs used for palliative purposes can be responsible for the occurrence of the syndrome. We mention the following: cyclophosphamide, vinorelbine, vincristine, cisplatin, ifosphamine, methotrexate, imatinib, non-steroidal anti-inflammatory drugs, opioids, some depressants and haloperidol. Even immunotherapy, through its immunosuppressive effects and the occurrence of immune-type disorders as adverse reactions can cause inflammation of the pituitary gland, increasing the level of the antidiuretic hormone [13].

Clinically, hyponatremia manifests with serum sodium levels below 120 mEq/L. The symptoms are represented by signs of water intoxication (hypoosmotic hyponatremia): apathy, confusion, convulsions and even coma. Focal neurological deficits may also be present [10].

The main therapeutic option for cases of SIADH is the treatment of the underlying disease and its correction depends almost exclusively on a good response to oncological therapy.

Other methods include fluid restriction, vasopressin 2 receptor antagonists – the class of drugs called vaptans (tolvaptan, conivaptan), 3% hypertonic saline solution, demeclocycline and lithium carbonate, the latter ones being used quite rarely [14].

In 2018, the Nobel prize for medicine was awarded to the doctors Jame Alisson and Tasuku Hojo for the description of PD-1 pathways and cytotoxic T-lymphocytes with antigen 4 (CTLA-4), showing their role in the inhibition of immune response checkpoints, which determined the improvement in the antitumor response to T-lymphocytes [15]. Ipilimumab, an anti CTLA-4 antibody, has proven effective in the treatment of malignant melanoma. [16]. Nivolumab, the first monoclonal antibody antagonist to PD-1, initially showed its effectiveness in non-small cell bronchopulmonary cancers and in melanoma and renal cancers, subsequently proving its contribution in other cancers, such as ENT cancers [17].

Another anti PD-1 agent, i.e., pembrolizumab, has proven its antitumor activity in cancers, such as non-small cell lung cancer, melanoma, ENT cancers and triple negative breast cancers [17,18,19,20].

Immunotherapy has revolutionized oncology and continues to do so through the emergence of newer and newer therapeutic agents.

## 2. Case Presentation

The second part of the paper, actually its main topic, represents a clinical case closely related to the presented theoretical notions.

This study is about a female patient who was 62 at the time of diagnosis who appeared in the records of the Oncology Department of “Sf. Luca” Chronic Disease Hospital in Bucharest. This female patient was diagnosed in February 2016 after a computed tomography of the head, thorax and abdomen with a right lung tumor, on which surgery was performed upon by practicing a right upper lobectomy with a mediastinal lymphadenectomy. The histopathological diagnosis indicated papillary lung adenocarcinoma, stage IIA – T2b N0 M0, with molecular markers (negative EGFR, ALK, PDL-1).

Postoperatively, it was decided to administer adjuvant chemotherapy in four series with carboplatin and paclitaxel. The follow-up tomography did not reveal any sign of a local tumor recurrence or other oncological suspicions. The patient remained under observation, undergoing regular check-ups every six months. Two years after the completion of the adjuvant chemotherapy, a follow-up tomography (July 2018) revealed confluent adenopathy in the aorto-pulmonary window and both the left hilar and left bronchial, which extended precarinarily, being indistinguishable from each other and up to 20/27 mm or 32/22 mm in size. A PET-CT that was performed revealing a metabolically active left mediastinal and hilar adenopathy.

The thoracic surgical examination revealed an inoperable left mediastinal-pulmonary relapse. The case was discussed in a multidisciplinary board and conformational or stereotactic radiotherapy was proposed but the patient declined this option.

Chemotherapy was resumed in four series of Gemcitabine and platinum salts. The follow-up tomography in January 2019 revealed the disease was in a slight progression. Some of the previous adenopathies were dimensionally reduced (in the aorto-pulmonary window, from 18 mm short axis to 14 mm short axis) but others had increased (left inferior bronchus from 12/16 mm to 22/27 mm), with a newly appeared adenopathy in the prevascular space of 14/17 mm (Figure 1).

Chemotherapy continued with Vinorelbine in monotherapy, administered weekly.

In June 2019, the CT scan revealed unfavorable aspects when compared to the previous examinations: an increase in the mediastinal mass to 50/42 mm, the increase in the mediastinal adenopathies, the occurrence of peribroncho-vascular iodophilic tissue densifications adjacent to the left inferior lobe and by the appearance of numerous nodules and micro nodules at the level of the left inferior lobe, lung metastases, hepatic cysts, a right renal cortical cyst, splenic hemangioma, osteocondensation at the level of the left iliac wing—to be correlated with a bone scan, which subsequently denied secondary bone determinations.

Immediately after the investigations and examinations, the patient was presented to the hospital with an altered general condition: confusional syndrome, nausea, vomiting, with an ECOG 4 performance status, and from a biohumoral point of view, with severe hyponatremia (Sodium 117 mmol/L with a normal inferior limit of 135 mmol/L), mild hypokalemia (K 3.3 mmol/L with a normal inferior limit of 3.5 mmol/L), hyperthyroidism (low TSH and increased T3 and T4). After the administration of sodium and potassium chloride, the hypokalemia remitted but the hyponatremia worsened, reaching 113 mmol/L. An endocrinology consultation was requested for the severe hyponatremia, and for the initiation of immunotherapeutic treatment for the metastatic disease. The patient was transferred to “C.I. Parhon” National Institute of Endocrinology, where the diagnosis of hyperthyroidisms was made in the context of polynodular goiter and paraneoplastic SIADH. The adrenal cortex failure was denied due to the increased basal cortisol level in the context of acute stress. She received recommendations for the treatment of hyperthyroidism; fluid restriction was recommended for the hyponatremia. The patient was given the approval to initiate oncological treatment, considering the fact that the first therapeutic intention in the occurrence of a paraneoplastic syndrome is the treatment of the underlying disease. 

Nivolumab was initiated for the progressive metastatic disease associated with SIADH with 240 mg being administered once every two weeks. The patient remained under observation, following the ionogram, so that at the next administration of immunotherapy the sodium and potassium levels were within normal limits. The patient was in a good general condition, without any complaints, with a ECOG 1 performance status. Practically, we found ourselves in front of a dramatic and extremely favorable and rapid response, monitoring the daily evolution of a patient who started at an ECOG 4, a situation which caused the severe hyponatremia and who reached an ECOG 1 state without obvious symptoms in just a few days, following a single administration of the immunotherapy treatment.

The subsequent evolution of the patient was clinically favorable, with a constant ECOG 1 and with a general condition that was without any significant complaints. The imagine examinations were repeated approximately once every six months. A CT scan was performed in 2020, revealing a right adrenal nodule, which required a differential diagnosis between adenoma and a secondary determination, which then required monitoring. Additionally, the CT scan revealed the regression of the micronodules and the lung nodules that were previously described and the regression of the mediastinal adenopathy.

Considering the good general condition, the following imaging investigation was expected in order to define the prognosis according to the RECIST criteria. It was performed in August 2021 and it revealed bilateral pulmonary micronodules to be monitored, mediastinal adenopathy (Figure 2), a hepatic lesion of the V segment (Figure 3) and bilateral adrenal nodules suspicious for secondary determinations (Figure 4).

The patient continued the treatment with Nivolumab considering the clinical benefit criterion, despite the suspicion of disease progression.

The next CT scan was performed in March 2022, which revealed dimensional progression and pulmonary secondary determinations (Figure 5), liver metastases (Figure 6) and adrenal secondary determinations, the occurrence of abdominal adenopathy (retroperitoneal and mesenteric) and newly occurring peritoneal carcinomatosis (Figure 7). 

It was the time to question whether to continue with immunotherapy or with a palliative treatment.

However, the Oncology Therapeutic Indication Board favored the clinical benefit against the imaging progression. The patient could move by herself, she continued with various light activities, she is conscious, cooperative and still comes for treatment almost a year after the previously mentioned decision.

It is worth mentioning that along the way, and also in the last six months from the date of writing this paper (April 2023), there have been two episodes of hyponatremia: the first in October 2022 was moderate hyponatremia with a sodium level of 124 mmol/L and the second in January 2023, also with a sodium level of 122 mmol/L, both of which were corrected after the administration of immunotherapy.

The next CT scan was performed in February 2023 revealed an aspect suggestive of oncological disease progression with the appearance of a nodule in the right cerebellar parenchyma suspicious of a secondary determination, which was to be completed with brain MRI (in the absence of contraindications) (Figure 8). Additionally, the CT scan revealed the numerical and dimensional progression of the secondary lung lesions (Figure 9), the marked dimensional progression of the secondary adrenal tumors, the right one invasive in the liver parenchyma, the progression of the secondary liver lesions (Figure 10), the progression of the mediastinal and abdominal adenopathies, the disappearance of the secondary splenic lesions and peritoneal carcinomatosis lesions and the dimensional regression of the secondary peritoneal lesion in the left hypochondrium (Figure 11).

The brain MRI performed in March 2023 did not reveal brain metastases.

## 3. Discussion

SIADH is a syndrome characterized by water retention associated with inadequate antidiuretic hormone secretion (ADH) [21]. This can appear as a paraneoplastic syndrome through an ectopic production of arginine Vasporesin peptide-like from cancer cells. It can appear through an endogenous hyperproduction of ADH from the pituitary gland or it can be iatrogenic due to a reaction from the administration of some drugs, including certain cytostatic drugs, such as cyclophosphamide, cisplatin, methotrexate and vinorelbine.

Generally, the time of onset of this syndrome or the remission time guides us in clarifying its etiology. Hyponatremia that develops gradually over a period of time is most likely the result of an endogenous production.

Hyponatremia is the most common complication of solid tumors having a negative impact on the quality of life in patients [21].

The discontinuation of some drugs that have this reaction associated with hyponatremia remission indicate an iatrogenic cause [21].

Moreover, the remission of the syndrome after specific interventions, such as surgery, radiotherapy and chemotherapy suggests an ectopic production of cancer cells, with a favorable response to the oncological treatment.

Obvious cases of hyponatremia remission after surgical interventions in ENT cancers have been reported [22].

A case with NCSLC was reported for the first time along with a good response to chemotherapy by normalizing the sodium levels immediately after the administration [23].

Severe SIADH is associated with an increased mortality rate among hospitalized patients. In those patients with obvious neurologic symptoms, such as cerebral edema require a rapid intervention in the first 48 hours. The objective is the increase in sodium level with 1–2 mmol/L through the administration of a 3% hypertonic saline solution. Special attention should be paid to not correct by more than 8–10 mmol/L in the first 24 h and 18–25 mmol/L in the first 48 hours [24].

Other therapeutic options are antagonists of V2 arginine vasopressin receptors (the vaptan class: tolvaptan, satavaptan), demeclocycline (300–600 mg × 2/day) and urea (30 g/day) [25].

Non-small cell lung cancer is a very rare cause of SIADH. Only a few cases have been highlighted in the specialty literature in the last decades. In a study conducted on 427 patients with NSCLC, only 0.7% presented this syndrome. [26]

The case presented in the current paper is a first in the specialty literature, given the following factors:SIADH in NSCLC (an extremely rare situation, 0.7%);Initiation of immunotherapy at the occurrence of severe hyponatremia, with its remission, and also the remission of two other episodes of hyponatremia (no other cases with obvious causal relationship between SIADG and a favorable response to immunotherapy have been reported);Long-term survival in the context of lung cancer diagnosed in 2016, metastasized in June 2019, the patient underwent immunotherapy with Nivolumab for three years and a half.

## 4. Conclusions

Lung cancer is the cancer with the highest incidence associated with the highest mortality rate in oncological pathology. This is one statement that triggers a high particular interest in the research of this disease. 

Although it cannot be considered statistically significant, it is important to also know rare situations, including their approach and evolution, such as the case presented in this paper: non-small cell lung cancer and SIADH, with a favorable response to specific immunotherapy treatment. 

The already proven success of immunotherapy is reinforced by the clinical evolution of this case with a much longer survival when compared to the normality of previous decades: lung cancer diagnosed seven years ago, which metastasized three and a half years ago and a living patient at the time of writing this paper (April 2023).

Another important aspect is the decision to continue with the oncological treatment, given the progression of the disease from an imaging point of view but with an obvious control of the hyponatremia, which is associated with a clinical benefit a decision that most probably led to the prolongation of survival versus the continuation of a palliative treatment.

The conclusion drawn is that each case must be approached individually, taking into account the various particular aspects of each patient.

Each newly emerged situation that is studied will definitely lead to an improvement in knowledge and to the increase in safety and progress, which will contribute to increase the general survival and to an improvement in the quality of life.

## Figures and Tables

**Figure 1 life-13-01279-f001:**
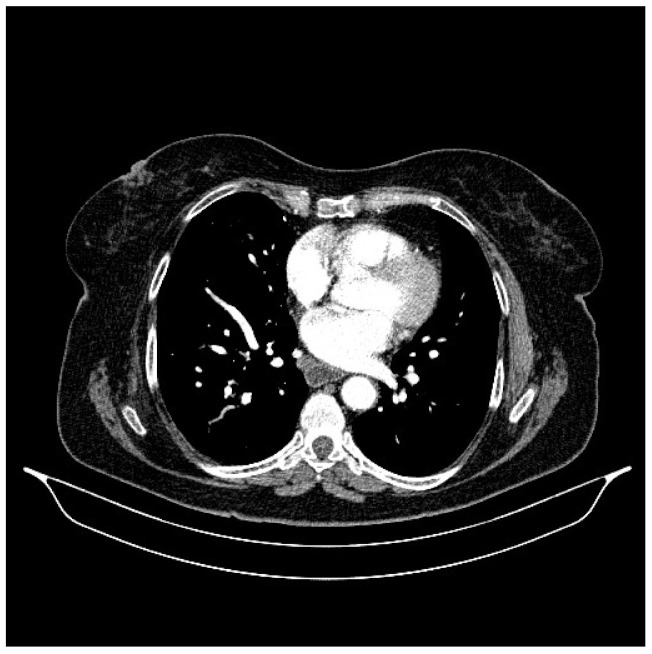
Chest CT, January 2019: some of the previous adenopathy’s were dimensionally reduced (in the aorto-pulmonary window, from 18 mm short axis to 14 mm short axis) but others had increased (left inferior bronchus, from 12/16 mm to 22/27 mm), with a newly appeared adenopathy in the prevascular space of 14/17 mm.

**Figure 2 life-13-01279-f002:**
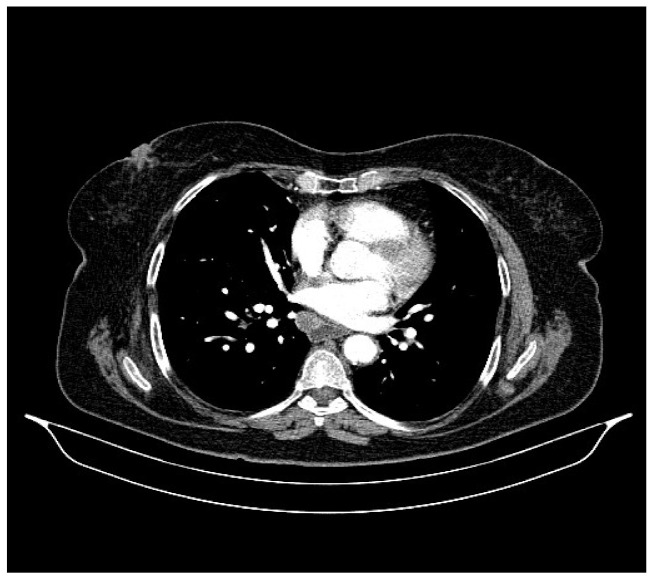
Chest CT, August 2021: bilateral pulmonary nodules to be monitored and mediastinal adenopathy.

**Figure 3 life-13-01279-f003:**
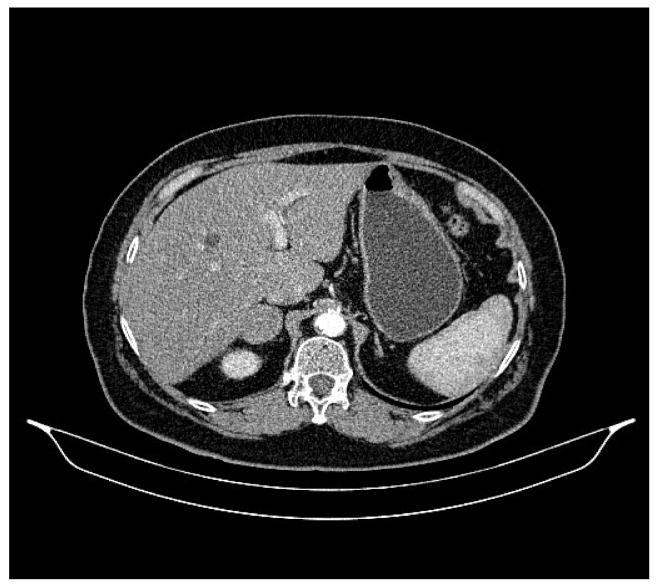
Abdominal CT, August 2021: a hepatic lesion of the V segment.

**Figure 4 life-13-01279-f004:**
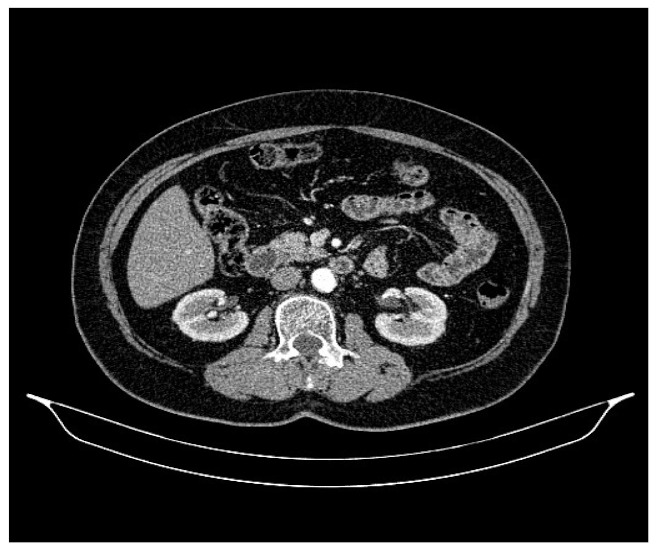
Abdominal CT, August 2021: bilateral adrenal nodules suspicious for secondary determinations.

**Figure 5 life-13-01279-f005:**
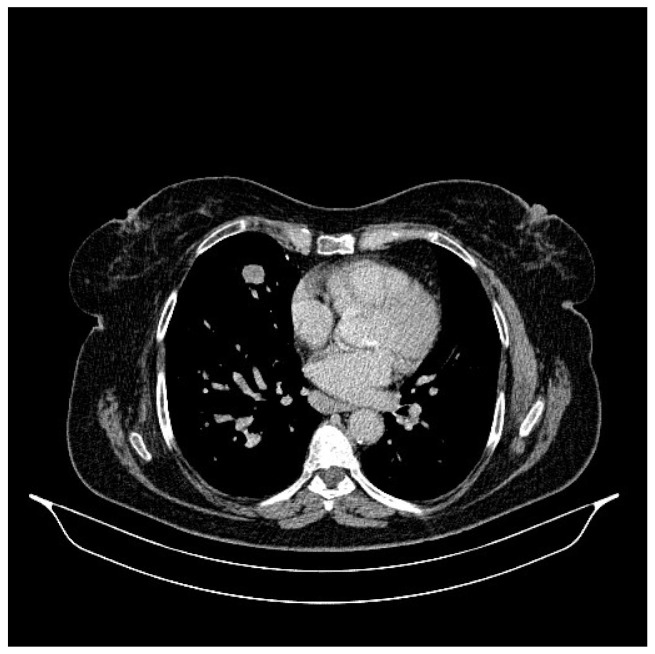
Chest CT, March 2022: dimensional progression and pulmonary secondary determinations.

**Figure 6 life-13-01279-f006:**
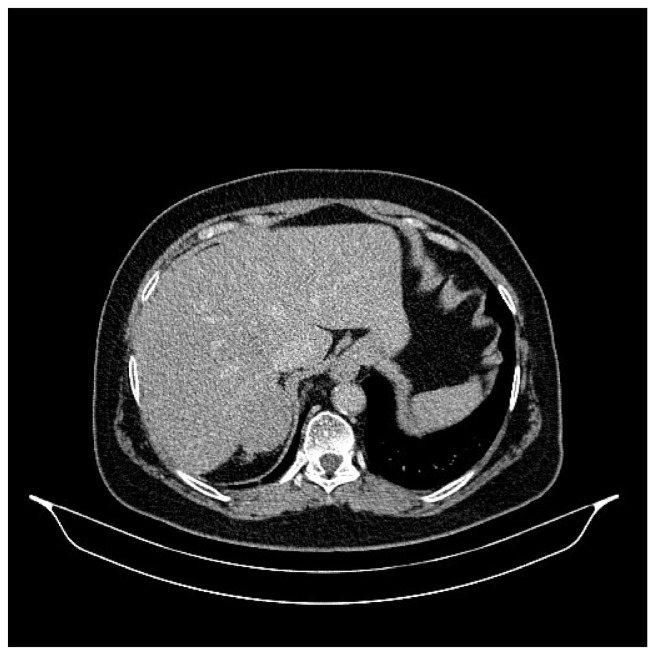
Abdominal CT, March 2022: liver metastases.

**Figure 7 life-13-01279-f007:**
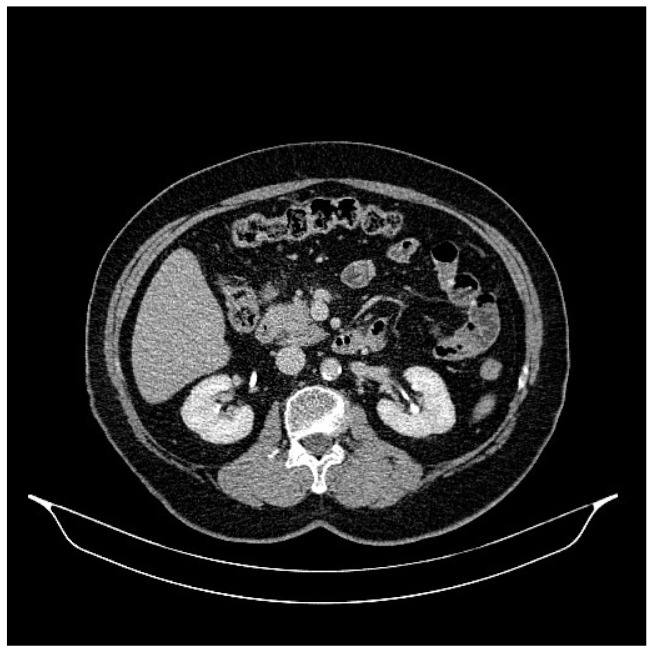
Abdominal CT, March 2022: adrenal secondary determinations, the occurrence of abdominal adenopathy (retroperitoneal and mesenteric) and newly occurring peritoneal carcinomatosis.

**Figure 8 life-13-01279-f008:**
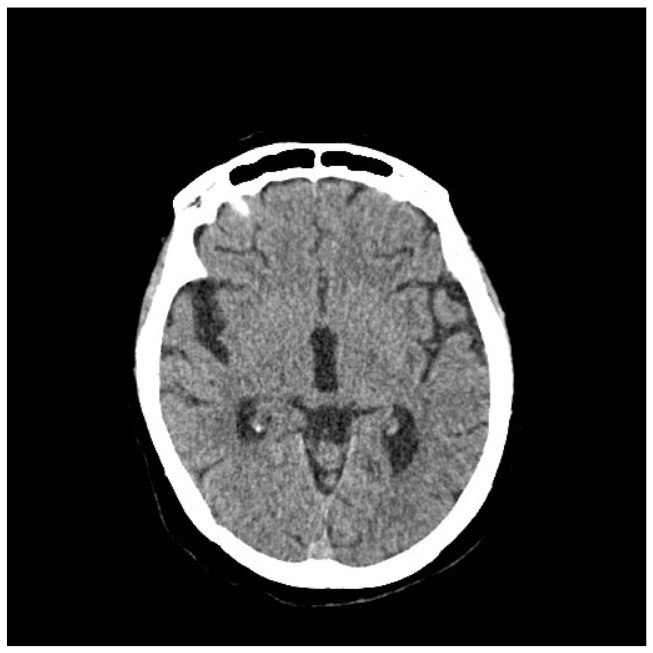
Brain CT, February 2023: appearance of a nodule in the right cerebellar parenchyma suspicious for a secondary lesion.

**Figure 9 life-13-01279-f009:**
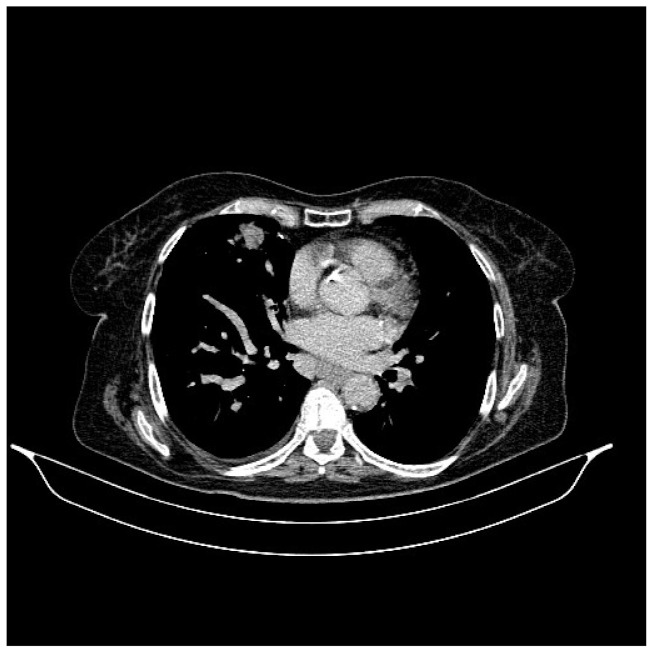
Chest CT, February 2023: numerical and dimensional progression of secondary lung lesions.

**Figure 10 life-13-01279-f010:**
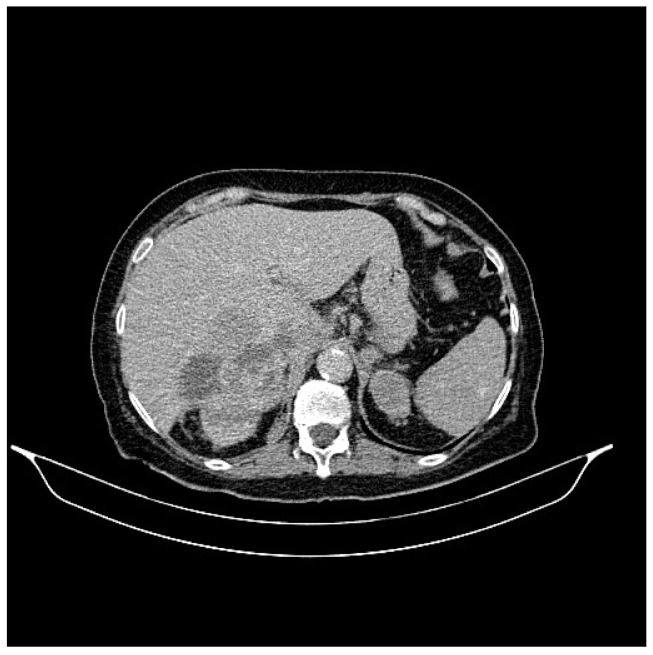
Abdominal CT, February 2023: progression of secondary liver lesions.

**Figure 11 life-13-01279-f011:**
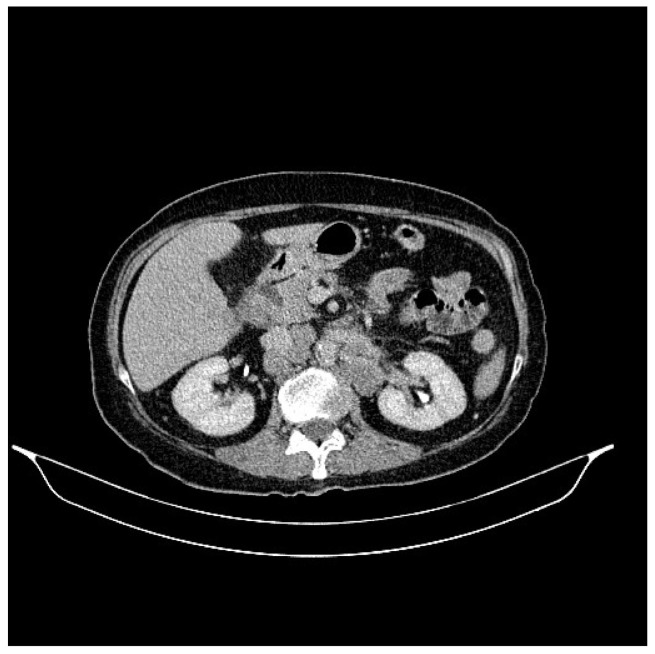
Abdominal CT, February 2023: disappearance of secondary splenic lesions and peritoneal carcinomatosis lesions, dimensional regression of the secondary peritoneal lesion in the left hypochondrium.

## Data Availability

Not applicable.

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
