# Peer review of "The Efficacy of Immunotherapy in Long-Term Survival in Non-Small Cell Lung Cancer (NSCLC) Associated with the Syndrome of Inappropriate Antidiuretic Hormone Secretion (SIADH)"

_life, 2023, doi:10.3390/life13061279_

Round 1

Reviewer 1 Report

SIADH often occurs in patients with small cell lung cancer, and the authors present a special case of SIADH occurring in non-small cell lung cancer. I have a few questions to answer. Does the patient have normal renal function before and after receiving immunotherapy? Does the patient have other routine tests in addition to imaging studies? Mechanistic exploration of this case will make the article more complete.

The author 's language is more fluent and the expression is more authentic, but there are some small details to pay attention to, and the article format should be more rigorous.

Author Response

Thank you so much for reviewing our work!

Our answers are:

-the patient had normal renal function before and after receiving immunotherapy

-the other routine tests that the patient had are the laboratory exams, that only revealed the hyponatremia, mild hypokalemia (but only in 2019, not at the first admission in hospital) and also hyperthyroidism (increased T3 and T4, low TSH)

-regarding the mechanistic approach, we added in the manuscript some physiopathological data that we found in the literature.

Thank you so much again for your time and for your help!

Reviewer 2 Report

life-2414518

In this case report study titled “The efficacy of immunotherapy in long-term survival in non-small cell lung cancer (NSCLC) associated with the syndrome of inappropriate antidiuretic hormone secretion (SIADH)” by Rahnea-Nita et al., the authors have presented a 62-year-old pulmonary adenocarcinoma patient clinical presentation after different treatment regimen including chemo and immunotherapy from the date of diagnosis in 2016 to April 2023. In this clinical case study, most of the discussions are focused on the efficacy of immunotherapy with regard to patient prognosis with a clinical presentation of syndrome of inappropriate antidiuretic hormone secretion (SIADH). Study is This manuscript discussed the case in an exploratory way but can gain from the following minor concerns.

As per DOs and DON’Ts of studies involving human patients, authors are responsible for obtaining written-informed consent and ethical approval. Provide details about Ethics Approval, declaration of Helsinki and informed patient consent before publishing the study.

With follow up Chest CT performed in 2019 provided in Figure 1, there is a reduction of some previous adenopathies but increase in left inferior bronchus adenopathy and a new adenopathy in the perivascular space. It would be better to have, if available chest CT from 2018 to compare the adenopathies during the course of treatment.

Authors mentioned that with endocrinology consultation the patient was diagnosed with hyperthyroidism and polynodular goiter and paraneoplastic SIADH. Patient received recommendations for the treatment of hyperthyroidism, and fluid restriction was recommended for hyponatremia. However, Nivolumab is initiated for the progressive metastatic disease, associated with SIADH, administered 240 mg once every two weeks. At the next administration of immunotherapy, the sodium and potassium levels were within normal limits, the patient being in a good general condition, without any complaints.

Question: Was patent under recommendations for the treatment of hyperthyroidism, and fluid restriction even during the treatment of Nivolumab. If yes, how can authors substantiate the effect of immunotherapy in correcting hyponatremia if patient was receiving hyperthyroidism treatment and was on fluid restriction. If not, what was the status of hyperthyroidism after immunotherapy treatment? It is worth to address these questions in manuscript as well for clarity.

Again, in lines 232-235, …. there have been two episodes of hyponatremia…. both being corrected after the administration of immunotherapy. Was the patient under fluid restriction at this point of time or immunotherapy alone? 

Authors mentioned that the patient was diagnosed with papillary lung adenocarcinoma, stage IIA – T2b N0 M0, with molecular markers negative for EGFR, ALK, PDL-1. However, patient was benefitted from anti-PD1 therapy Nivolumab.

Was there any mutational profiling (e.g., Foundation One) requested for this patient? If yes, I would recommend authors to show-case mutational signatures that may provide an idea about how PDL-1 negative tumors would benefit from anti-PD1 therapy.

Finally, there are a lot of grammatical and typo errors throughout the manuscript. For example, line 174-175 where the diagnosis of hyperthyroidisms in made in the context…. I would strongly recommend thorough standard English proof reading of the manuscript.

There are a lot of grammatical and typo errors throughout the manuscript. For example, line 174-175 where the diagnosis of hyperthyroidisms in made in the context…. I would strongly recommend thorough standard English proof reading of the manuscript.

Author Response

Hello and thank you for your advices!

  1. We attached the approval of the Ethical Committee
  2. CT from 2018 is described in our work (lines 136-138)

Tomography (July 2018) revealed confluent adenopathy in the aorto-pulmonary window, left hilar and left bronchial, extended precarinarily, indistinguishable from each other, of up to 20/27 mm or 32/22 mm in size

CT from 2019 (lines 145-149)

Tomography in January 2019 revealing the disease in slight progression, some of the pre-vious adenopathies being dimensionally reduced (in the aorto-pulmonary window, from 18 mm short axis to 14 mm short axis), but others increase (left inferior bronchus, from 12/16 mm to 22/27 mm), with newly appeared adenopathy in the prevascular space of 14/17 mm (Figure 1).

We do not have the film from 2018, but, from the written interpretation , we can observe the decrease of the dimensions of some adenopathies, in 2019, comparating with the aspect from 2018

Our patient has done a lot of tomographies in these 8 years, it was impossible to put all of them in the paper, so we chose some of them and we considered that the written interpretation is relevant.

Thank you so much for your attention at the details in our work, but from 2018 we only have the written interpretation.

  1. Regarding the patient’s treatment for hyperthyroidism, she was under treatment with beta-blockers and no restriction fluid at home for hyponatremia. The tyroid function was within normal limits. So, taking into account these facts and that the correction of hyponatremia happened everytime after administration of Nivolumab, we deduced that it was its role.

4.Regarding the indication for starting immunotherapy with nivolumab for progression metastatic disease, it isn’t neccessary a PDL-1 pozitive patient. We completed a questionnaire which has criteria for inclusion and in it, it is written: ”lung cancer, other than small cell lung cancer, metastatic, with histopathological confirmed”. Mollecular status of the patient was EGFR, ALK, PDL-1 negative, but this wasn’t a contraindication for starting the treatment.

We reattached the manuscript corrected by a translator. We also attached the consent of the patient (no 487/16.01.2023).

Thank you again for your time and for your valuable advise!

Reviewer 3 Report

The paper describes a well known paraneoplastic syndrome in Lung Cancer, that of SIADH, as well as an established treatment for Lung Cancer, that of Immunotherapy. It is interesting though as a case study, the representation of exact improvement in sodium levels, post immunotherapy, as well as the various staging scans. It would be interesting to know about previous lines of treatment, whether the patient was having treatment with PPIs or ACE inhibitors that are known causes of hyponatraemia, as well as any other diuretics and whether sodium levels were checked via a 24h urine collection. It is a useful addition to the literature for the use of immunotherapy and management of this complicated paraneoplastic syndrome.

Author Response

Thank you for reviewing our work!

Our answers are:

-the previous lines of treatment included chemotherapy with Gemcitabine and platinum salts

-the patient did not receive ACE inhibitors

-we did not check sodium levels via a 24h urine collection.

Thank you so much again for your time and for your help!

Round 2

Reviewer 1 Report

After the authors' revision of the manuscript, I have no other comments and I think the manuscript can be published.

Author Response

Thank you so much!